# Lipid-mediated hydrophobic gating in the BK potassium channel

**Lucia Coronel** [1], **Giovanni Di Muccio** [2,3] ✉, **Brad S. Rothberg**[4], **Alberto Giacomello** [2] **& Vincenzo Carnevale** [1] ✉

Structures of the large-conductance, calcium-activated potassium (BK) channel in the $Ca^{2+}$–bound and $Ca^{2+}$–free states have suggested that $K^+$ conduction is not gated via a steric closure of the pore-lining helices of the channel, in contrast to the gating mechanism of other 6TM channels. This has raised the question of how gating might occur in the absence of apparent steric hindrance by protein residues. To answer this question, we perform molecular simulations and free-energy calculations to develop a microscopic picture of the gating mechanism. Our results highlight an unexpected role for annular lipids, which appear to be an integral part of the gating machinery. In the $Ca^{2+}$–free ("closed") pore, methyl groups from lipid alkyl chains can enter the pore through fenestrations between the pore-lining helices. This dynamic occupancy directly contributes to dewetting of the inner-pore cavity, thus hindering ion conduction. In contrast, $Ca^{2+}$ binding leads to occlusion of the fenestrations, thus preventing the lipids from entering the pore cavity and permitting pore hydration and ion conduction. This apparent lipid-mediated hydrophobic gating may also explain functional observations that include state-dependent pore accessibility of hydrophobic channel blockers.

Large-conductance, calcium-activated potassium channels (BK channels) are finely tuned molecular machines that gate cellular $K^+$ efflux in response to cytosolic $Ca^{2+}$ binding and depolarization. They are ubiquitously expressed in nerves and muscles and control various biological functions ranging from hearing and neurosecretion to smooth muscle contraction.

Each subunit of the tetrameric BK channel consists of a voltage sensing domain (VSD), a pore-gate domain (PGD), and a cytosolic tail domain (CTD) (Fig. 1). The PGD enables permeation of selected ionic species across the cell membrane, while the VSD and CTD allosterically modulate pore gating. The transmembrane region of each monomer comprises seven helices, S0 through S6, with S1–S4 responsible for detecting membrane depolarization (VSD) and S5–S6 constituting the PGD. The loop between S5 and S6 forms the selectivity filter (SF), which accommodates at least two $K^+$ ions

simultaneously[1] (Fig. 1b). A water-filled central cavity in the pore is also the binding site for some cationic pore blockers[2]. The CTD contains binding sites for $Ca^{2+}$, and $Ca^{2+}$ binding triggers channel opening. BK channels can be activated independently via either $Ca^{2+}$ binding to the CTD or membrane depolarization, which activates the VSD to trigger channel opening[3,4]. An interesting property of BK PGD, compared to other potassium channels, is a pronounced kink of the pore-lining S6 helix between Ala316 and Ser317[5]. A rotation at this kink is part of the conformational transition that underlies in channel gating[6–8]. Even though several experimental structures in WT $Ca^{2+}$–bound (open) and WT $Ca^{2+}$–free (presumed closed) states have elucidated mechanisms of these structural modules, several crucial aspects of the gating mechanism remain unclear. In particular, although the WT $Ca^{2+}$–free conformation (PDB 5TJI[9]) presents a slightly narrower pore radius compared to the WT $Ca^{2+}$–bound one

[1]Institute for Computational Molecular Science and Institute for Genomics and Evolutionary Medicine and Department of Biology, Temple University, Philadelphia, PA, USA. [2]Department of Mechanical and Aerospace Engineering, Sapienza University of Rome, Rome, Italy. [3]NY-Marche Structural Biology Center, Department of Life and Environmental Sciences, Marche Polytechnic University, Ancona, Italy. [4]Department of Medical Genetics and Molecular Biochemistry, Temple University Lewis Katz School of Medicine, Philadelphia, PA, USA. ✉e-mail: g.dimuccio@univpm.it; vincenzo.carnevale@temple.edu

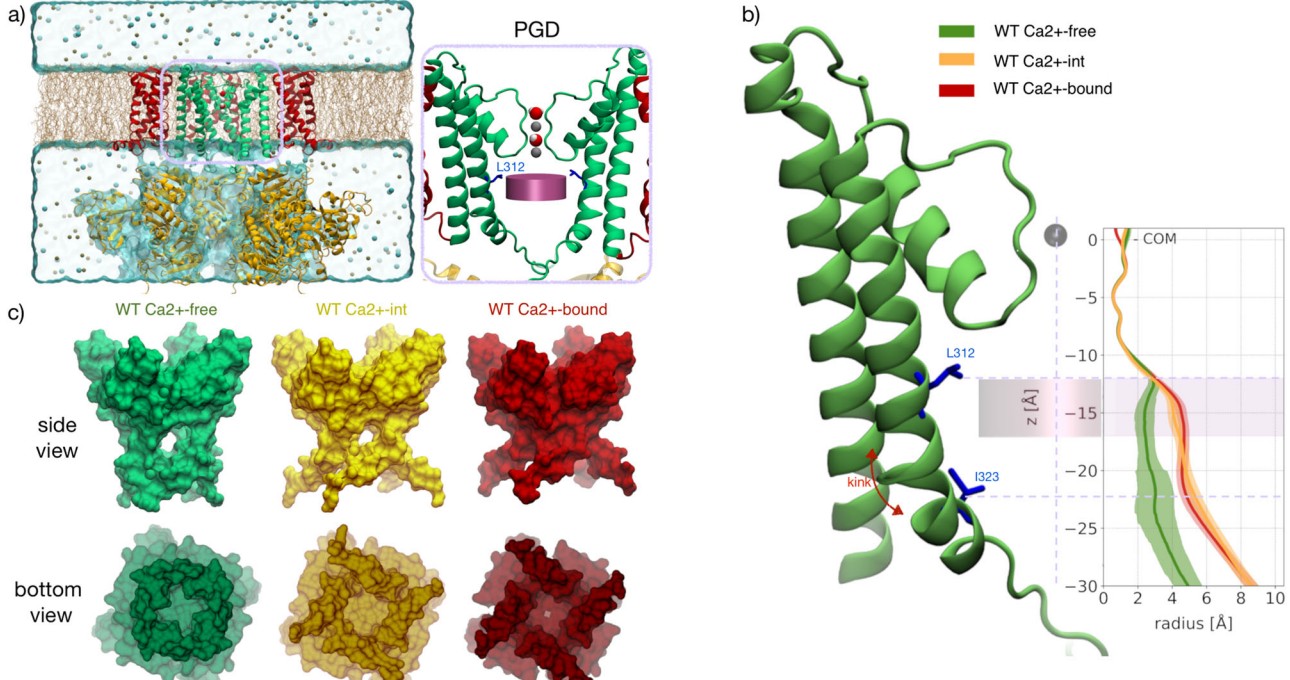

**Fig. 1 | Conformation and radius profile of the WT Ca²⁺−free, WT Ca²⁺−int, and WT Ca²⁺−bound states. a** Side view of BK channel. Protein in cartoon representation: the S1-S4 helices (Voltage-Sensing Domain, VSD) are highlighted in red, the S5–S6 helices in green (Pore-Gate Domain, PGD), and the intracellular portion (C-Terminus Domain, CTD) in yellow. Membrane lipids (ochre) and ions are shown as thick lines and spheres, respectively. The water molecules in the selectivity filter are shown in space filling representation. For clarity, only three subunits are shown. In the inset, the PGD is shown (only two subunits shown for clarity) along with the K⁺ ions (gray spheres) and waters occupying the Selectivity-Filter (SF).

The deep pore volume (DPV) and the side chain of residue L312 are shown as pink cylinder and blue sticks, respectively. **b** Average radius profiles of the WT Ca²⁺−free (green), WT Ca²⁺−int (yellow), and WT Ca²⁺−bound (red) states (pdb codes 6V3G, 8GHG, and 6V38, respectively). z = 0 is set at the SF center of mass and the pink shaded area indicates the location of the DPV. **c** Surface representation of the Cryo-EM structures (side and bottom views): WT Ca²⁺−free (green, pdb code: 6V3G), WT Ca²⁺−int (yellow, pdb code: 8GHG), and WT Ca²⁺−bound (red, pdb code: 6V38). Error bars represent standard deviations.

(PDB 5TJ6[9]), neither structure shows a pore that is sufficiently occluded by the pore-lining helices to hinder diffusion of waters and ions to and from the central cavity. In the WT Ca²⁺−free conformation, the radius of the pore is in principle still large enough (as large as ∼6 Å) to allow occupancy by multiple water layers, ions and other molecules[9–12]. One of the molecular mechanisms that has been proposed to explain the transition toward a non-conductive state is hydrophobic gating[11,13]. This mechanism relies on dewetting phenomena that can occur within narrow hydrophobic pores or channels, owing to a combination of spatial restriction and unfavorable interaction between water molecules and exposed non-polar residues[14–19]. Since the BK inner PGD core presents a hydrophobic region just below the SF, in particular near residue Leu-312, the hypothesis that hydrophobic gating may occur in BK channels is supported by several computational and experimental studies[11–13]. This mechanism may explain how the pore could remain accessible to relatively large hydrophobic molecules while relatively small ions become impermeant. However, the BK PGD volume seems to exceed the critical radius, length, and hydrophobicity needed for the "bubble nucleation" that disrupts ion permeation[13,16,19–23]. According to the model developed by Rao et al.[20] to predict the presence of a hydrophobic gate, the WT Ca²⁺−free BK channel structure is hydrated. This suggests that some other, thus far unidentified molecular events may contribute to the dewetting of the BK channel pore.

Here we explore the molecular basis of pore-gating in the BK channel via a quantitative investigation of WT Ca²⁺−bound and WT Ca²⁺−free structures through extensive Molecular Dynamics (MD) simulations that combine equilibrium and enhanced sampling calculations. Our analysis unveils additional physical and thermodynamic features underlying the gating process and strongly emphasizes the fundamental, yet unanticipated, role of membrane lipids in the dewetting transition. BK gating may depend on lipids that partially breach the conduction pathway through lateral fenestrations, gaps between the S6 helices that line the pore cavity[24,25], in the WT Ca²⁺−free conformation. The intrusion of lipids into the cavity results in steric hindrance and increases the hydrophobic character of the pore inner cavity, triggering dewetting. Finally, these annular lipids may also play a crucial role in the coupling between the PGD and CTD; Ca²⁺ binding to the CTD causes an apparent displacement of lipid molecules that makes them incapable of permeating the fenestrations and inhibiting ion conduction. Overall, our results reveal unexpected regulatory mechanisms governing BK channel activation that rely on the interplay between Ca²⁺ binding, PGD-lipid interactions, and hydrophobic gating.

## Results

### Pore hydration and lipid penetration

To gain insight into the structural basis of BK channel gating, we collected an extensive set of 1 μs-long MD trajectories of the human BK channel, based on cryo-EM structures of WT Ca²⁺−bound and WT Ca²⁺−free states−PDB accession numbers 6V38 (WT Ca²⁺−bound) and 6V3G (WT Ca²⁺−free)[26]. Additionally, we studied a structure of the BK channel acquired in EDTA-free solution but with no Ca²⁺ added (PDB accession number 8GHG)[27]. Since in the absence of EDTA free Ca²⁺ is present as a contaminant of KCl and other salts at a concentration of ≈20 μM, we refer to the 8GHG structure as "intermediate" (hereafter denoted as WT Ca²⁺−int). The WT Ca²⁺−int structure was previously observed to exhibit C2 molecular symmetry, only slightly different

from the C4 symmetry of the WT $Ca^{2+}$ −bound structure, consistent with the idea that this conformation represents BK channels in which the $Ca^{2+}$ binding sites are less occupied[27]. Thus, the WT $Ca^{2+}$ −int conformation may be intermediate between the fully WT $Ca^{2+}$ −bound (open) state and the WT $Ca^{2+}$ −free (presumed closed) state.

Using measurements of protein atoms in the WT BK channel structures, the WT $Ca^{2+}$ −bound and WT $Ca^{2+}$ −free states show distinct radius profiles in the region delimited on one end by the cytosolic mouth and on the other by $Ca^{2+}$ −int structure has a pore lumen radius that resembles the WT $Ca^{2+}$ −bound structure (Fig. 1b). The pore radius of the WT $Ca^{2+}$ −free configuration, although small, is twice the radius of a water molecule, i.e., this section is large enough to accommodate two to four water molecules. Thus, although the presence of multiple hydrophobic residues within the DPV potentially contributes to dewetting, it may not be sufficient to trigger a "bubble nucleation"[11]. Although the pore radii of WT $Ca^{2+}$ −bound, WT $Ca^{2+}$ −int, and WT $Ca^{2+}$ −free BK channels are all seemingly large enough to accommodate water and ions, inspection of our molecular simulation trajectories reveals qualitatively different behaviors. Specifically, the DPV of the WT $Ca^{2+}$ −free state evolves from a wet to a dewetted state during the course of the simulation, consistent with previously reported observations[11,28]. Notably, we find that the dewetting event occurs specifically when methyl groups of annular lipid tails, breach the DPV through fenestrations, partially occupying the pore lumen (Fig. 2d). To quantify the extent of these events, we monitored the number of lipid carbon atoms and water molecules within the DPV over time (Fig. 2a).

Despite the fact that the initial configuration of our systems shows no lipids tails breaching these fenestrations, after 200 ns the WT $Ca^{2+}$ −free DPV is breached and becomes occupied by up to 8 ± 3

carbon atoms. In contrast, in the WT $Ca^{2+}$ −bound and WT $Ca^{2+}$ −int states, the hydration level remains approximately constant along the trajectory with a vanishing lipid occupancy of the pore. The difference between the conformations is summarized by the distributions of water and lipids counts computed after 200 ns: the water average counts are 0.8, 14.1 and 15.6 for the WT $Ca^{2+}$ −free, WT $Ca^{2+}$ −bound, and WT $Ca^{2+}$ −int, respectively, while the average lipid counts are 7.7 for WT $Ca^{2+}$-free and 0 for the others (Fig. 2a).

Steric hindrance by lipid carbon atoms occupying the DPV was quantified by computing the DPV radius, taking into account the protein structure and the lipids. Compared to the WT $Ca^{2+}$ −bound state, the radius of the WT $Ca^{2+}$ −free DPV (at z ∼ −15 Å) is decreased by 40%, passing from $r_p$ = 2.4 Å to $r_{p+l}$ = 1.5 Å (Fig. 2b-pink line). Intrusion of lipid carbon atoms in the WT $Ca^{2+}$ −free structure is possible due to four symmetry-related fenestrations connecting the pore lumen to the central region of the lipid bilayer. These fenestrations present in the WT $Ca^{2+}$ −free structures arise from the kinked conformation of S6 (Figs. 1c and 2e). Although the kink is present also in the WT $Ca^{2+}$ −bound and WT $Ca^{2+}$ −int structures, only the WT $Ca^{2+}$ −free undergoes dewetting. The S6 tilt angle distributions of WT $Ca^{2+}$ −free, WT $Ca^{2+}$ −bound, and WT $Ca^{2+}$ −int indicate that the latter two states adopt a more bent conformation, as reflected by their larger angles, resulting in narrower or completely closed fenestrations. These results are, in general, consistent with the observation of electron density near the fenestrations attributed to annular lipids in cryo-em structures (e.g., in PDB ID 6V38)[9,12,26,29]. Strikingly, a recent high-resolution cryo-EM structure of the WT $Ca^{2+}$ −free state of the BK channel in lipid bilayers revealed the presence of lipid molecules breaching the fenestrations and occupying the pore lumen in a conformation that is

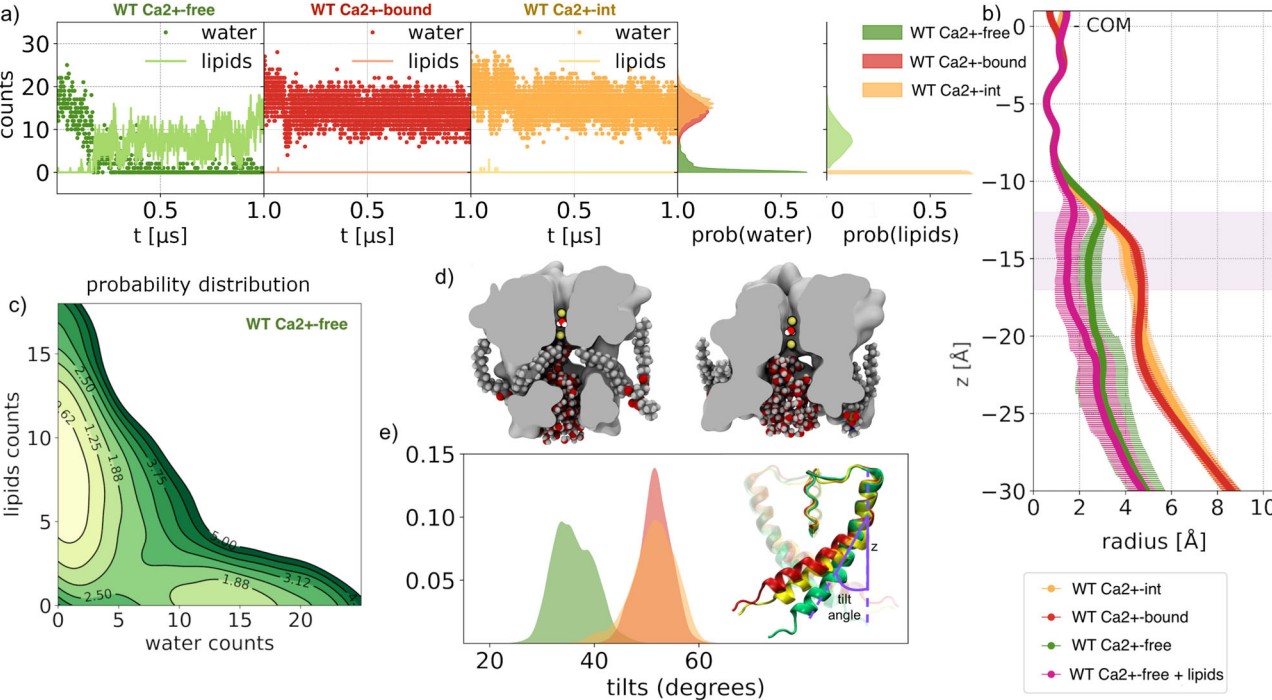

**Fig. 2 | Pore hydration and lipid penetration. a** Lipid carbon atoms and water molecules count within the DPV as a function of time and their relative probability densities computed for t ≥ 200 ns. **b** Average radius profiles: WT $Ca^{2+}$ −free, WT $Ca^{2+}$ −int, and WT $Ca^{2+}$ −bound conformations are shown in green, yellow, and red, respectively. The pink line shows the radius profile of the WT $Ca^{2+}$ −free conformation when lipid atoms are included in the calculation. z = 0 corresponds to the SF center of mass, and the pink shaded area indicates the location of the DPV. Error bars represent standard deviations. **c** The two-dimensional histograms of water and lipid atom counts for the WT $Ca^{2+}$ −free conformation. For visual clarity, a negative

logarithmic scale is used for plotting the values obtained through kernel density estimation and the global minimum is set to zero. **d** Molecular surface of the WT $Ca^{2+}$ −free and WT $Ca^{2+}$ −bound conformations sampled from the MD trajectories. To highlight the shape of fenestrations, the location of the intruding lipids and the pore occupancy by water molecules, a section of the PGD is shown along the C4 axis of symmetry. **e** Comparison of S6-helix tilt angle probability distributions. In the inset, the tilt angle (defined as the angle between S6 helix and vertical axis) is shown for the WT $Ca^{2+}$ −free (green), WT $Ca^{2+}$ −int (yellow), and WT $Ca^{2+}$ −bound (red) states.

completely analogous to the one observed in our simulations[30]. Overall, water and lipid pore occupancies are mutually exclusive, as suggested by the anticorrelation shown in Fig. 2c, which highlights two distinct DPV states: a hydrated one without lipids, and a dewetted one with lipids.

Dewetting was observed in 2 out of 3 of the 1 μs WT Ca²⁺ -free trajectories (Fig. S2a), whereas it was never observed in the two WT Ca²⁺ −int ones (Fig. S2c) or in the two WT Ca²⁺ −bound ones (Fig. S2b). To further investigate the differences between WT Ca²⁺ −free and WT Ca²⁺ −int, we performed metadynamics simulations and quantitatively characterized the stability of each lipid occupancy state.

We collected 0.7 μs-long trajectories of WT Ca²⁺ −free and WT Ca²⁺ −int states using the lipid carbon atom density in a sphere as collective variable, as described in Methods. Both trajectories show anticorrelation between hydration levels and lipid counts (Fig. 3a). However, the lipid-count free-energy profiles are significantly different (Fig. 3b). The WT Ca²⁺ −free free energy profile shows two local minima, separated by a minor barrier of approximately 1–2 kcal/mol. Interestingly, the state corresponding to a high lipid count (≈35) is the energetically favorable one. In contrast, the free-energy profile of the WT Ca²⁺ −int state has the global minimum at low lipid density (≈10), consistent with a greater hydration level favoring ion conduction.

## Metastability of hydrated and dewetted state

Since the transition between the wet and dry state of a nanocavity can be considered a rare event[31], estimating quantitatively the relative probabilities of these states requires sampling over extended time scales. Hence, to establish how the stability of the two states varies with the number of lipid carbon atoms present in the DPV, we employed enhanced restrained MD simulations[19,32].

Using this approach (see Methods for details), we calculated hydration free-energy profiles as a function of the number of water molecules within the DPV, for various structures extracted from the WT Ca²⁺ −free and WT Ca²⁺ −int trajectories. For each sampled structure, in the absence of intruding lipids, no metastable minima are observed in the dewetted state (Figs. 4d and S8). However, when lipid intrusion occurs, hydration free-energy profiles reveal two minima, corresponding to hydrated and dewetted (meta)stable states (Fig. 4). Specifically, Fig. 4 highlights how increasing the number of lipids within the cavity influences the relative stability of the hydrated and dewetted state: for a lower number of lipid carbon atoms, the dry state is stable (replica r1, r2); as the number reaches $n_L$ ~7, the two states become equiprobable (r3, r4); for larger values, the dry state becomes the most probable (r5, r6).

In the dry state, water molecules are absent from the region in contact with the potassium ions within the SF. This absence is noteworthy since ions must cross this region during conduction[33] and is in line with the previously proposed hydrophobic gating mechanism[11]. Importantly, no such dewetting is observed unless lipids are present in the pore: this is true for both the WT Ca²⁺ −free structure, as well as the WT Ca²⁺ −int structure. This indispensable role of lipids in triggering dewetting does not depend on the ions configurations in the SF (we examined both S1−S3 and S2−S4 configurations) or the force-field adopted for the simulations (the restrained MD simulations were performed using both the SPC/Eb and TIP3P water models): see Fig. S8d and Supplementary Figs. S8 and S9.

Altogether, our results support the conclusion that the intrusion of lipid tails into the DPV is a critical component of the dewetting transition to a non-conductive configuration. This conclusion is consistent with observations reported in previous studies[28,34], where a dynamic occupancy of the pore lumen by lipid tail was deemed responsible for the "lack of convergence" of hydration free energy profiles. Rather than preventing lipid diffusion into the pore as was done in these previous investigations, here we systematically investigated the effect of lipid occupancy on hydration metastability and clarified that, depending on their penetration depth, lipids can shift the equilibrium from the hydrated to the dewetted state.

## Deep-pore mutations that stabilize the open state prevent dewetting

It was previously observed that mutations at residue L312, which is located within the DPV near the site of lipid-breaching (Fig. 1), can have substantial effects on BK channel gating energetics. In patch clamp experiments, the L312A mutant shows a median voltage of activation close to 0 mV even at nominally 0 Ca²⁺, representing a shift of around −180 mV compared to WT. The L312D mutation has an even stronger impact, resulting in an apparently constitutively open channel[7]. To determine whether these gating effects may be related to the energetics of lipid-breaching or dewetting, we performed MD simulations by incorporating the mutations in the context of the Ca²⁺ −free BK channel structure and analyzed pore radii and hydration levels of L312A and L312D mutants.

Remarkably, we have observed that both L312A and L312D mutations result in the transition of the Ca²⁺ −free conformation into one similar to the wild-type WT Ca²⁺ −bound, displaying similar values of the pore radius in the DPV (Fig. 5c) and hydration levels (Fig. 5a). In addition, we observe that during the course of the trajectories, lipids do not breach the DPV sufficiently to induce dewetting. Specifically, the L312A pore radius in the DPV (light-blue profile) closely resembles

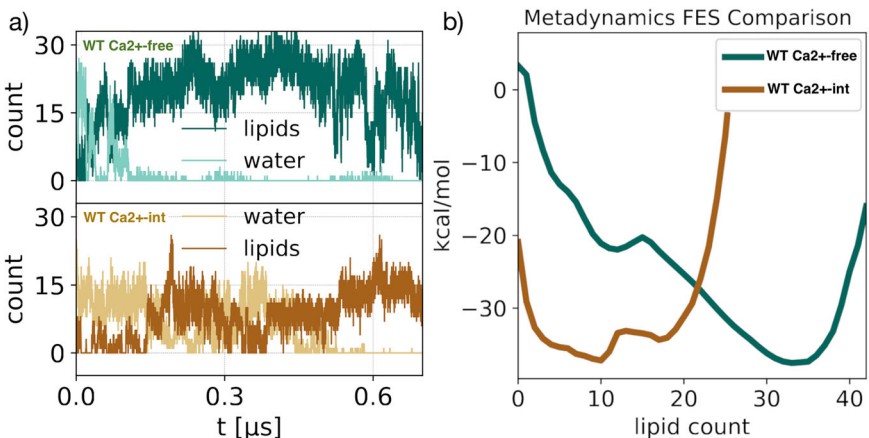

**Fig. 3 | State-dependent lipid affinity of the pore.** Metadynamics trajectories collected for WT Ca²⁺ −free (green) and WT Ca²⁺ −int (yellow)states. **a** Lipid carbon atoms and water molecules counts within the DPV. **b** Free-energy profiles as a function of lipid density.

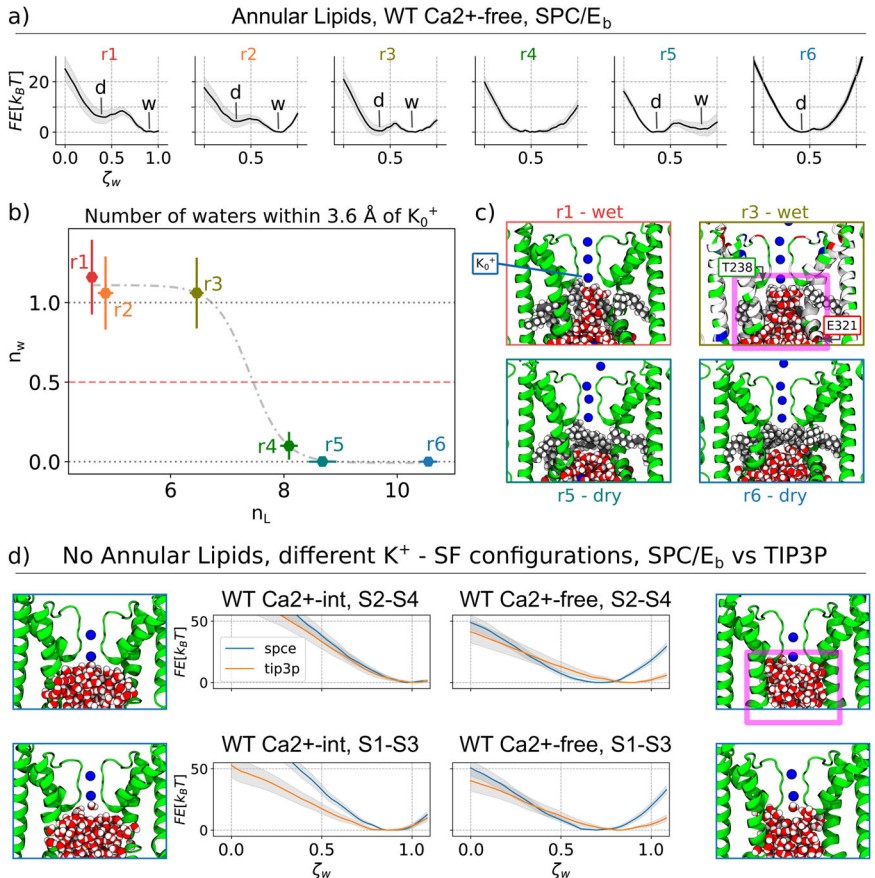

**Fig. 4 | Hydration free energy profiles for different pore replicas.** Each replica has a distinct configuration and a different number of lipids inside the lumen. The free energy profiles are obtained by thermodynamic integration of multiple Restrained Molecular Dynamics (RMD) simulations using as a collective variable the number of water molecules inside a control box surrounding the hydrophobic region of the pore lumen, indicated in magenta in **c**, top right. **a** The profiles are ordered by decreasing number of lipid carbons inside the control box, i.e., replica r1 has the largest number of lipids and r6 the lowest. Calculations are performed using SPC/Eb water model in combination with Amber force field. A comparison with TIP3P water model is shown in Supp. Fig. S9, showing a similar switching trend. Gray shaded areas represent standard deviations. **b** Number of water molecules within 3.6 Å of the innermost potassium in the SF (here indicated as K0+) for each global minimum of the profiles of (**a**). This number (here referred to as nw) reports

on the amount of water molecules in direct contact with the SF in the equilibrium state. Error bars represent standard deviations. **c** Molecular configurations sampled from the global minimum of replicas r1, r3, r5, r6. The pictures clearly show that water molecules are in contact with the SF in r1 and r3 but not in r5 and r6. Top views are also shown in Supp. Fig. S12. **d** Free energy profiles computed in absence of lipids, for the WT $Ca^{2+}$-int and WT $Ca^{2+}$-free states. Plots show the comparison between two force-fields (CHARMM36-TIP3P vs Amber-SPC/Eb) and two SF ions configurations (ions were kept constrained in two different configurations, S1-S3 and S2-S4). Overall, to assess the robustness of our predictions, a total of 8 additional free-energy profiles were calculated on the truncated pore model (see Methods for details). Grey shaded areas represent standard deviations. Consistent free-energy profiles calculated on the full-length protein model are reported in Supp. Fig. S10.

that of the WT $Ca^{2+}$-bound state, while converging toward the WT $Ca^{2+}$-free (green) values of 5 − 6 Å at the cytosolic mouth. The L312D pore radius (blue) is greater than that of WT $Ca^{2+}$-bound. The increased DPV radius in both mutants, compared to the WT $Ca^{2+}$-free conformation, results in an increase in water molecule count, with an average values of $18 \pm 3$ and $21 \pm 3$ for L312A and L312D respectively, while the lipid count remains <6.

The L312A and L312D mutations also impact the S6 helix rearrangement (Fig. 5b) by increasing the S6 tilt angles, thereby reducing the fenestration size. Specifically, the distribution of helix angles for both mutants shifts toward the WT $Ca^{2+}$-bound state values, although this effect is less pronounced than that of the WT $Ca^{2+}$-int state (Fig. 2e). With the L312D mutant, the distribution is bimodal, an indication that not all four helices tilt to the same extent.

Even though the L312A mutation preserves the DPV's hydrophobic character the pore remains hydrated (Fig. 5a) and permeable to ions throughout the simulations, providing further support for the idea that breaching of lipids through the fenestrations are a critical step to dewetting.

## Potential involvement of lipids in allosteric coupling

In addition to producing a significant leftward shift of the voltage-activation curve, several side chain substitutions at position 312 greatly reduce apparent $Ca^{2+}$ sensitivity (i.e., the V1/2 of the G-V curve with 0 $Ca^{2+}$ is nearly the same as that obtained at 85 μM $Ca^{2+}$ [7]). Therefore, we wondered if annular lipid molecules involved in pore dewetting might also be involved in allosteric coupling between the CTD and PGD[3,7,12,35–38]. To test this, we examined potential interactions between the CTD and PGD that might be mediated by lipids (Fig. 2d) (Fig. S5). For the fenestration-penetrating lipid, we observed consistent polar head group interactions of with Arg329 and Lys392 side chains (Figs. 6a, b and S5). While Arg329 and Lys392 interact with the lipid head groups in both the WT $Ca^{2+}$-free and WT $Ca^{2+}$-bound states, it is only in the WT $Ca^{2+}$-free state that acyl tails from the interacting lipids are close enough to the fenestrations to penetrate the pore lumen. Specifically, during the course of our 1 μs-long trajectories, we observed that the lipids whose phosphate groups were at a distance of less than 6 Å from R329 had terminal methyl groups inside the pore in the WT $Ca^{2+}$-free but not in the WT $Ca^{2+}$-bound state (Fig. 6c, d). This result is consistent with the idea that the sequence [329]RKK[331] is

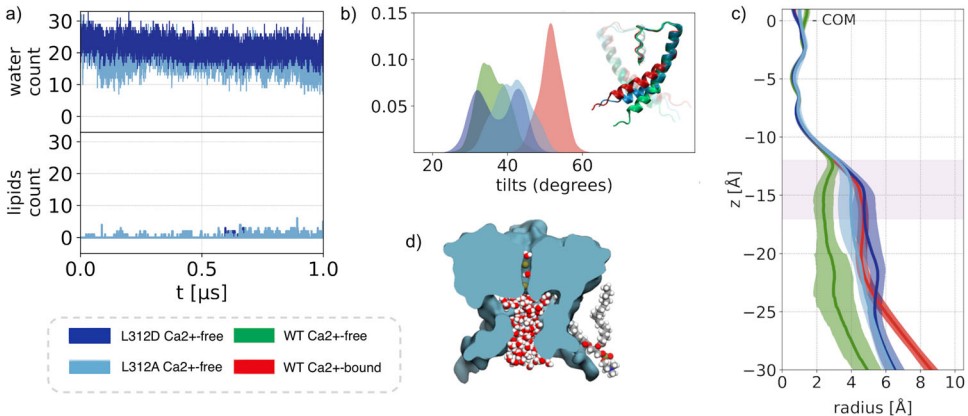

**Fig. 5 | Hydration properties of deep-pore mutants.** L312A and L312D Ca²⁺−free trajectories: **a** lipid carbon atoms and water molecules counts within the DPV. **b** S6-helix tilt angle probability density functions. In the inset, a comparison is shown between the S6 helix conformation of the WT Ca²⁺−free (green), WT Ca²⁺−bound (red) states, and L312A (blue). **c** Average radius profile of the WT Ca²⁺−free (green), WT Ca²⁺−bound (green), and L312A,D mutants. z = 0 is set at the selectivity filter center of mass and the pink shaded area indicates the location of the DPV. Shaded areas represent standard deviations. **d** Molecular surface of L312A showing the water filled central cavity as well as one annular lipid.

important to channel opening[39]. Remarkably, during our simulations, only the the lipid tails containing the unsaturated bond (the oleic acid ones) were able to penetrate the fenestrations. This preference for the oleoyl moiety, which possibly results from the "kinked" configuration of the unsaturated alkyl chain, raises the possibility that BK gating may be finely regulated by the structural properties of the surrounding lipid environment, consistent with what previously experimentally observed[40]. Overall, in addition to the rotation and translation of the S6 helices that lead to the closing of fenestrations (Fig. S3), Ca²⁺ binding causes a ~9 Å movement of Lys392 from the αB helix (Fig. S6) that may prevent the associated lipid from breaching the fenestration (Fig. 6a, b).

## Discussion

In this work we investigated the structural basis of BK channel gating using equilibrium and enhanced sampling molecular dynamics simulations. Our findings align with the view that BK channel gating is controlled by the hydration level of the central cavity, as this region is differentially hydrated in the WT Ca²⁺−free and WT Ca²⁺−bound states. Notably, it is within this zone that dewetting occurs, due to the penetration of lipid tail alkyl chains, which can stably occupy the fenestrations in simulations.

The notable occurrence of lipid breaching in the WT Ca²⁺−free BK channel conformation highlights the critical role of annular lipids in facilitating the dewetting transition, which in turn precludes ion permeability. Our findings suggest that the stability of the dewetted state depends on the level of intrusion of the lipid tails into the cavity. We determined that below a certain threshold of carbon atoms inside the DPV ($n_L = 7$), both the dewetted and hydrated states are equally probable, but a higher number of lipids tips the balance in favor of the dewetted state; conversely, a lower number favors the hydrated state. Interactions between lipids and specific amino acids within the S6 helix C-terminus and the C-terminus domain (CTD) appear vital for favoring the penetration of lipids and for keeping the lipid heads close to the central cavity. This supports the notion of a dynamic, functional connection between lipid-CTD interactions and channel activity.

The structure and stability of the fenestrations appears to be a key factor determining pore dewetting, and thus gating energetics. It was previously observed that mutations at residue L312 (in the fenestrations region) can elicit hyperpolarizing shifts of the BK channel voltage-activation curve, consistent with a stabilization of the conductive state. In the WT Ca²⁺−free conformation, mutations L312A and L312D, which are experimentally observed to stabilize the conductive

state, result in narrowing of fenestrations, and thus blocking of lipid-breaching of the pore. Additional experiments will be required to further understand the impact of other pore residues on fenestration dynamics and their impact on channel gating.

Our analysis has revealed an unexpected potential role of lipids on pore hydration and its impact on BK channel gating, as well as a potential mechanism to couple Ca²⁺ binding to the CTD and gating of the PGD. This active and direct role of lipids in gating paves the way for future studies and suggests potential approaches to modulate BK activity.

## Methods

### MD setup and equilibrium simulations of the BK channel

The BK channel is studied in the WT Ca²⁺−bound and WT Ca²⁺−free (pdb id: 6V38 and 6V3G, respectively[26]), as well as WT Ca²⁺−free and ETDA-free state (pdb id: 8GHG[27]). The molecular systems are assembled using the CHARMM-GUI membrane builder web server[41–47]. Specifically the channels are embedded in a lipid bilayer of 1-palmitoyl-2-oleoyl-sn-glycero-3-phosphocholine (POPC) with a number of ions adjusted to obtain electrical neutrality at a salt concentration of 0.15 M. Molecular dynamics simulations are performed by keeping the temperature and pressure constant via the Nosé−Hoover temperature coupling method (with a time constant of 1 ps), and semi-isotropic Parrinello−Rahman method (with a time constant of 5 ps), respectively[48–51]. The long-range contribution of electrostatic interactions resulting from periodic boundaries conditions is calculated using the Particle-Mesh-Ewald algorithm[52]. Two potassium ions and two water molecules are located in the selectivity filter, in the S1-S3 and S2-S4 binding sites, respectively. This arrangement is kept constant by applying harmonic restraints to the ions and waters positions by using PLUMED 2.8[53]. Note that these are the only restraints enforced in MD simulations. The system's interactions are described by the CHARMM36 force-field[43,44,54] with TIP3P water[55]. Equilibrium simulations are carried out using GROMACS 2021.4 and PLUMED-2.8[53,56–61]. After an initial minimization and equilibration of ~2 ns, following the standard CHARMM-GUI protocol, the system is evolved using a 2 fs timestep and collecting trajectories for a total of 1000 ns. In total, we collected three independent trajectories for the WT Ca²⁺−free conformation, and two for the WT Ca²⁺−bound and WT Ca²⁺−int conformations. Each of these 7 molecular systems are independently generated by CHARMM-GUI. The results obtained for the replicas are shown in the Supplementary Information. We use HOLE software[62] to calculate the pore radius. For each conformation, we calculated the

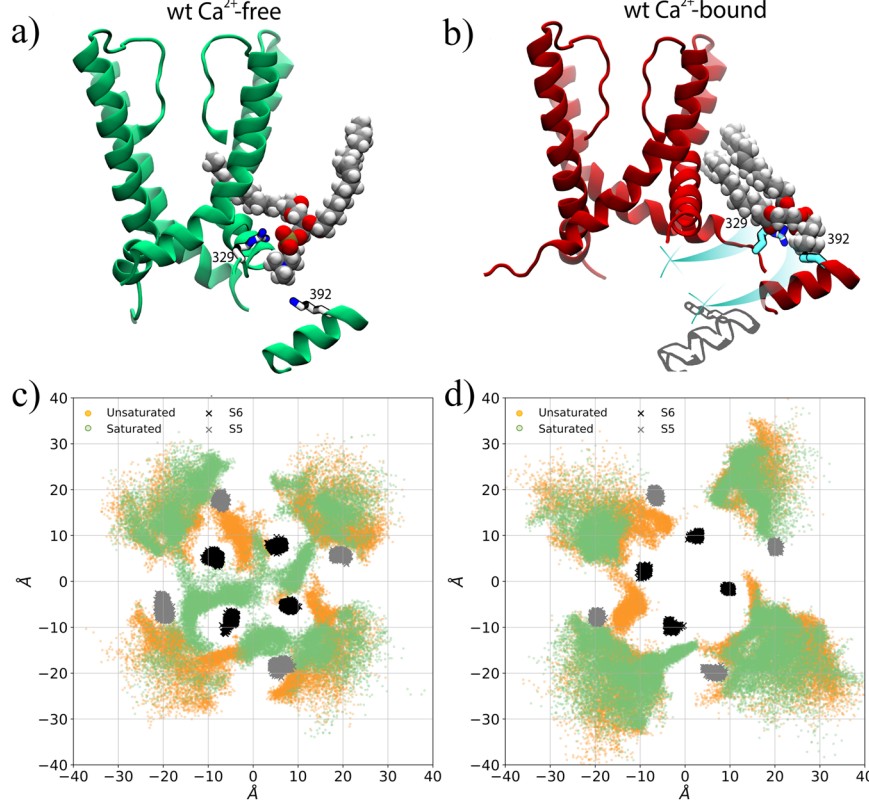

**Fig. 6 | Coupled motions of αB helices and annular lipids. a, b** Molecular configurations of the **a** WT Ca²⁺–free (green) and **b** WT Ca²⁺–bound (red) conformations. Lys392 and Arg329 are shown as sticks, while the lipids that are simultaneously within 6 Å of S6, Lys392, and Arg329 are shown in VDW representation. Light blue lines indicate the displacements of Lys392 and Arg329

residues between the two states. **c, d** Projection on the xy plane of the position of the annular lipid atoms along with S5 (black) and S6 (grey) for the **c** WT Ca²⁺–free and **d** WT Ca²⁺–bound conformations. The positions of the atoms from the hydrophobic tail of the lipid are shown in green (oleic acid, unsaturated) and orange (palmitic acid, saturated).

average pore radius and determined the error through block analysis. We designed a specific region of interest within the deep pore, hereafter referred to as deep pore volume (DPV), as follows: a cylindrical volume with a radius of 0.7 nm and axis aligned to the SF axis of symmetry. The cylinder's height is 0.5 nm with its topmost circular face located 1.2 nm below the SF center of mass and the other in proximity of the kink of S6 (Fig. 1). The number and type of molecules contained within this region is estimated using VMD[63] scripts, while the S6 helices orientation is studied using MDAnalysis HELANAL-routine[64,65]. Mutants of L312 are generated using the CHARMM-GUI web server[66], simulated with the same protocol used for the wild-type structures. Metadynamics simulations, carried out using PLUMED[53,59–61], are collected with 2 fs timestep, for a total of 700 ns each. As collective variable, we use the lipid tail carbon atoms density, ρ, in a spherical volume of radius 0.7 nm, centered in the COM of the four symmetry-related residues Phe315. Free energy profiles are computed by reweighting the configurations sampled via the biased metadynamics simulation. Error estimates are determined with block analysis.

**Enhanced RMD simulations and hydration free energy profiles**
From the equilibrium MD, we extract a series of configurations to better sample the wet/dry transition of the pore, in the presence or absence of annular lipids. For each extracted configuration, we use restrained molecular dynamics simulations (RMD)[67] to extract the hydration free energy profile of the pore, using NAMD[68] and the Colvars module[69]. For each configuration, the protein backbone and the phosphorus of POPC molecule heads are constrained. Simulations are performed using different water models: SPC/Eb[70] and TIP3P[55]. The two models produce different values of the surface tension[71], a

fundamental parameter in bubble nucleation and thus hydrophobic gating[16]. To assess the robustness of our result, some of the simulations are replicated using both models, see Figs. S8 and S9. SPC/Eb water is used in combination with the Amber force-field (ff15ipq force-field[72] for the protein and the Lipid17 force-field[73] for the phospholipids), while TIP3P is used in combination with CHARMM36 force-field[54]. RMD is conducted by restraining the number of water molecules inside a confined region of the space, by adding an harmonic potential to the Hamiltonian of the system

$$H_N(\boldsymbol{r},\boldsymbol{p}) = H_0(\boldsymbol{r},\boldsymbol{p}) + \frac{k}{2}\left(N - \widetilde{N}(\boldsymbol{r})\right)^2 \tag{1}$$

where $\boldsymbol{r}$ and $\boldsymbol{p}$ are the positions and momenta of all the atoms, respectively, $H_0$ is the unrestrained Hamiltonian, $k$ is a harmonic constant which is set to 1 kcal/mol, $N$ is the desired number of water molecules in the control box, and $\widetilde{N}$ is the related counter for the actual number of water molecules in the system at each step. A Fermi parameter equal to 3 Å is used to smooth the borders of the box and make the collective variable continuous. The protocol is implemented in NAMD by using the Volumetric map-based variables of the Colvars Module[69], as introduced in ref. 74. The center and size of the counting box is set equal to the center and the minmax size of the rings composed by the residues Thr287 and Glu321, computed by VMD[63]. In these calculations, we consider a region larger than the hydrophobic cavity, to avoid artifacts induced by the boundaries. The water molecules affected by the counting box are represented in VDW spheres in Fig. 4. Each filling state corresponding to a single point of the reported profiles in Figs. S8 and 4 is simulated for 4 ns.

Averages are performed after discarding the first transient phase of 2 ns; representative traces reporting the number of water molecules inside the control box along the simulation are reported in Fig. S7. Each trajectory is saved every 20 ps, while the number of water molecules inside the box is saved every 1 ps. The free energy gradient (the mean force) for each point is computed as $-k\langle \tilde{N} - N \rangle$[67]; standard error is computed via block average, with a block length of 100 ps. Finally, the free energy profiles are obtained by thermodynamic integration of multiple RMD simulations (40 points). Integration is performed using the Euler method; the standard errors are propagated correspondingly.

### Reporting summary

Further information on research design is available in the Nature Portfolio Reporting Summary linked to this article.

## Data availability

All data supporting the findings of this study are available within the article and its Supplementary Information. Data has been deposited in Zenodo [https://doi.org/10.5281/zenodo.15732171].

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

## Acknowledgements

This research includes calculations carried out on HPC resources supported in part by the National Science Foundation through major research instrumentation grant number 1625061 and by the US Army Research Laboratory under contract number W911NF-16-2-0189 and by EuroHPC PRACE through the awarded project ElectroHG on the LUMI@CSC infrastructure. V.C. acknowledges support by the National Institute for General Medical Science through grant 3R01GM093290. B.S.R. acknowledges support by the National Institute for General Medical Science through grants R01GM126581 and R01GM150218. This project has received funding from the European Research Council (ERC) under the European Union's Horizon 2020 research and innovation programme (grant agreement No 803213).

## Author contributions

B.S.R., A.G. and V.C. were responsible for the overall project design, direction and supervision. L.C. and G.D.M. conducted the simulations. All authors contributed to the discussion of results and paper writing.

## Competing interests

The authors declare no competing interests.
