## [Transparent Peer Review file · Nature Communications]

Lipid-mediated hydrophobic gating in the BK potassium channel

Corresponding Author: Professor Vincenzo Carnevale

Version 0:

Reviewer comments:

Reviewer #1

(Remarks to the Author)

This is an interesting work where atomistic simulations were applied to study the role of lipid fenestration in hydrophobic gating of BK channels. Hydrophobic gating has been proposed to be involved in BK gating in several previous studies. The major idea put forward in this work is that lipid fenestration, enabled by Ca^{2+} unbinding-associated pore conformational change, is crucial to hydrophobic gating, but not a byproduct or unessential contributor. This conclusion is based on the analysis of correlation between lipid occupancy and hydration free energy summarized in Figure 4. While this is an attractive proposal, I have reservation on its validity because of concerns on the technical details as well as contrasting observations from multiple published studies. The rest of the study is quite thin. As such, I have a difficult time finding enough strengths in this work to reach a favorable recommendation for its publication in Nature Communications.

1) About half of the paper is dedicated to analysis of water and ion distributions in Ca bound and free states. These properties are well documented in multiple studies at this point, including Jia et al 2018, Gu and de Groot 2023, Nordquist et al 2023, and Contreras et al 2024 (<https://doi.org/10.1101/2023.12.29.573674>).

2) Lipid fenestration in Ca-free BK channels is also well documented in both the above-mentioned simulation works, as well as a few published cryo-EM structures. The observation at this point from these studies is that lipid fenestration is inconsequential, as the pore was observed to be fully capable of dewet even without lipids in the pore. For example, a previous calculation of the pore hydration free energy profiles (Nordquist BJ 2023) show that the presence of lipid tail is not required for dewetting transition (in this work, restrains were imposed to exclude lipids from the inner cavity to achieve good convergence on the hydration free energy profiles).

It is hard to speculate why the results reported in the current manuscript contradict all these previous simulations. One probably has to carefully inspect the structure integrity of the channel used in free energy calculations. One notable observation is that the human BK channel structure 6v3g was used in the simulation for Ca-free state. This structure contains a spurious pi-bulge in pore-lining helices. This feature is not present in any other BK structures reported so far; it is not consistent with McKinnon's own cyro-EM density map either. Nordquist et al examined this feature in depth. It was observed that 6v3g can be unstable in simulations and the pi bulge will dissolve back into regular alpha-helix after long relaxation.

I think it is imperative that the authors carefully examine the structures used in their calculations and reexamine the requirement of lipid fenestration in dewetting transitions.

3) The definition of the DPV is potentially problematic as it only includes space where the lipid tail(s) would occupy. As such, it could exaggerate the correlation between pore hydration and lipid occupancy. Instead, one should define the entire inner pore volume, between the selectivity filter and cytosolic entrance, to examine how pore hydration depends on lipid occupancy

4) I am not sure the number of water near the bottom of the filter is a terribly meaningful indicator of the conductivity. Instead, it is the total number of water in the pore (or the level of dewetting in the entire pore) that is the most accurate indicator of the conductive state of the pore.

5) the assertion that “the BK PGD volume seems to exceed the critical radius, length and hydrophobicity demonstrated to be needed to allow and maintain a bubble nucleation required to disrupt ion permeation” in the introduction does not seem to be correct. Previous studies of carbon nanotubes or idealized hydrophobic nanopores suggest a critical diameter of ~1 nm, which also depends on length of the pore and actual hydrophobicity. It is difficult to anticipate what “the critical radius, length and hydrophobicity” would be for a stable dewet BK pore. Nonetheless, the dewet inner pore of BK appears to be no greater than ~6Å in diameter (see Fig 2 or in previous simulation works (Jia et al 2018 and Gu and de Groot 2023).

6) the statement “Indeed, despite previous studies have shown a correlation between the dehydration level of the pore and protein structural features, e.g., the orientation of Phe residues in the central cavity or the radius shrinking [22], the complete dewetting of the region just below the SF was rarely reported” is confusing

Reviewer #2

(Remarks to the Author)

This manuscript uses MD simulations to study the gating mechanism of MK potassium channels, addressing the question of how these channels are gated in the absence of a pore occluding bundle crossing gate. They find that lipids penetrating through the lateral fenestrations are critical for gating, with the presence of lipid tails in the pore dehydrating the pore cavity (but not occluding it) preventing ion conduction. The result is genuinely interesting and intriguing, presenting a very unusual mechanism of ion channel gating and as such makes this content suitable for Nature Communications. But, it is essential to make sure the observation is statically significant prior to publication. As such I have a few suggestions for improvement.

Major concerns

1. The most interesting finding to me is that lipid tails in the pore generate pore dehydration which prevents ion conduction. Furthermore lipid tails only enter the pore in the Ca²⁺ free WT conformation, but not in the Ca²⁺ bound conformation or the Ca-free L312A or L312D mutants. Given the centrality of this results, I would like a more complete explanation as to why lipid tails don't enter the pore, as it is not obvious for the L312A mutant for example. In that case, the mutation is very subtle and would not be expected to occlude the fenestration, not does it have a big influence on hydrophobicity. SO why don't lipid tails enter in this case?
2. Following on from this, the lipid and water occupancy is determined from single 1 us simulations. It looks like to me that the authors only conducted one simulation for each condition (w/wo Ca²⁺) for 1 us, in order to obtain statistical significance to show the solid difference observed between simulations, the authors must conduct multiple replicas. The authors can also make better use of the multiple fenestrations in each simulation that yield pseudo replicates that can better determine the uncertainty in the lipid occupancies.
3. Finally, how much is the lipid occupancy dependent on the starting configurations of the lipids? The authors make use of charm-gui to prepare the systems but this doesn't do particularly well in placing lipids around the transmembrane protein with complex conformation like BK. If doing multiple replicates, I would suggest making use of multiple starting lipid coordinates as well.
4. I would like additional justification for the choice of starting cryo-EM structures 6V38/6V3G which have relatively poor resolution, while there have been other BK channel structures in cell-derived membrane environment available (8GHF, 8GHG).
5. It is stated that the fenestration are not present in the Ca²⁺ bound conformation. Can you give some analysis to support this? This can be done on the cryo-EM structures and ideally could include all the available BK cryo-EM structures to show it is consistent across structures with the same Ca²⁺ state.
6. The RMD results shown in Fig 4 are interesting. But I would also like to see ana analysis of the pore radius in each case to see if the lipids are occluding the pore or dehydrating it, as these correspond to distinct gating mechanisms.

Minor comments:

7. When citing the early papers on hydrophobic gating I recommend you cite one of the first examples in biological channels: Biophys. J. 90: 799-810, 2006. I also recommend citing prior examples of lipid induced gating eg Nature Commun (2022) 13:490
8. Were any restraints used to keep the backbone restrained - and to keep ca²⁺ free and Ca²⁺ bound conformations distinct? None are mentioned, but I just want to be sure.
9. It is stated that water/ion in SF are restrained. Why is this done and is it just for equilibration or in the production runs? This is particularly important as the authors were discussing the ion occupancy later, the occupancy of ion in SF has big impact of ion occupancy in pore cavity.
10. For the pore radii profile, please make it clearer when this is made using protein only and protein+lipid

11. Fig 3B – I recommend making the labels clearer to indicate that the grey lines are water occupancy and the colours are lipid. For example the labels could be the same colour as the line and the flat colours lines could be made more visible (eg thicker lines)

12. Please provide a reference when saying “These results are consistent with the experimental observation that the G-V curve of L321A in absence of Ca²⁺ is significantly left-shifted whereas L312D lead to constitutively open channel”.

Reviewer #3

(Remarks to the Author)

The authors describe a computational study that proposes a lipid mediated hydrophobic gating in the BK potassium channel. Hydrophobic/dewetting gating transitions have been proposed before in BK channels but the lipid involvement is for the first time systematically investigated here. Although in principle interesting, I have a number of concerns that should be addressed before I can recommend the manuscript for publication.

First, one of the main conclusions of the work "starting from a Ca²⁺-free leads to ion conduction upon Ca²⁺ binding." and "Ca²⁺ binding [...] allowing for pore hydration and conduction." is not backed up by the data, as no conductance has been demonstrated. The authors should show sustained currents in the calcium bound form consistent with experiments to support this claim.

Second, the presumed conductive state remains unclear in the current manuscript. In Figures 1 and 3, the selectivity filter is depicted as containing both water and ions in its binding sites. This is consistent with the description in the methods section: "Two potassium ions and two water molecules are allocated in the selectivity filter (TVGYG), in position S1-S3 and S2-S4 respectively.". However, in Fig. 4 filter configurations are shown containing 3 ions and no waters. The conductance for ion-only permeation might be quite different from a mechanism of ion/water co-permeation, so this should be clarified.

Third, it is not explicitly stated which calcium parameters were used. Classical parameters that come with the employed force fields have been shown to show significant overbinding for divalent cations. Hence, specific multisite calcium parameters have been developed to mitigate this issue. Were these used in the current study?

Finally, the manuscript suffers from numerous minor language issues and erroneous or missing literature references. To name just a few examples:

-the charmm36 reference is inaccurate

-Gromacs and plumed were mentioned but not cited

-"The two model have reproduce differently"

-"starting from a Ca²⁺-free leads"  "starting from a Ca²⁺-free state leads"?

-"we do not observed"

Version 1:

Reviewer comments:

Reviewer #1

(Remarks to the Author)

While I appreciate the efforts of the authors in responding to the concerns, I am unconvinced that my main concerns have been adequately addressed (reliability of the simulation, such as due to artifacts and instability problem of 6V38/6V3G; contrasting observations about the requirement of lipid fenestration for dehydration; limited new insights beyond the proposed role of lipids).

Reviewer #2

(Remarks to the Author)

The authors have made considerable effort to address the comments in the first review including conducting a number of new simulations and analysis. I believe they have responded well to the comments I raised in my initial review. While lipid entry to the fenestrations and pore dewetting are both topics that have been discussed previously, the clear link between the presence of lipids in the fenestration and pore dewetting described here is novel and interesting in my opinion.

Reviewer #3

(Remarks to the Author)

The authors have satisfactorily addressed my concerns

Lipid-mediated hydrophobic gating in the BK potassium channel

Reviewers' comments are in black

Authors responses are in blue

Reviewer #1 (Remarks to the Author):

This is an interesting work where atomistic simulations were applied to study the role of lipid fenestration in hydrophobic gating of BK channels. Hydrophobic gating has been proposed to be involved in BK gating in several previous studies. The major idea put forward in this work is that lipid fenestration, enabled by Ca²⁺ unbinding-associated pore conformational change, is crucial to hydrophobic gating, but not a byproduct or unessential contributor.

Thank you for your helpful feedback. We appreciate your interest in our work and your detailed comments. Indeed, our major claim is that lipid intrusion into the deep-pore volume (DPV) through lateral fenestrations is essential for the hydrophobic gating of the BK pore.

This conclusion is based on the analysis of correlation between lipid occupancy and hydration free energy summarized in Figure 4.

Thank you for your observation. While Figure 4 indeed provides a detailed analysis of the correlation between lipid occupancy and hydration free energy, our conclusions are supported by a comprehensive analysis of multiple simulated systems. Specifically, we examined four full-length structural models—including wild-type (WT) open and closed states and mutants—as well as six truncated configurations extracted from the full-length simulations. These results are presented in multiple figures (previously Figures 2 and 3) and are thoroughly described in the main text and Supplementary Figures S7 and S8. Altogether, our findings demonstrate that in the absence of lipids, the pore remains consistently hydrated.

While this is an attractive proposal, I have reservation on its validity because of concerns on the technical details as well as contrasting observations from multiple published studies.

Thank you for your thoughtful critique. We appreciate your reservations regarding the technical details and the contrasting observations from multiple published studies.

As you mentioned, previous simulation studies by Jia et al. (2018), Nordquist et al., and Gu & De Groot (2023) have reported the presence of lipids in the BK channel simulations. However, the role of these lipids was not systematically investigated in relation to hydrophobic gating. In Jia et al. (2018), occurrences of lipid intrusion were noted in the supplementary materials, but their impact on the gating process was not explored.

In the studies by Gu & De Groot (2023) and Nordquist et al., simulations included restraints that explicitly prevented lipid intrusion into the pore. Consequently, Gu & De Groot (2023) did not observe complete dewetting—likely due to these artificial restraints—and reported that a minimal amount of water was always present in the channel's cavity. Nordquist et al. observed complete dewetting; however, while lipid molecules were not excluded from the fenestration cavities, they were prevented from occupying the inner region of the cavity. This setup allowed for a decrease in the deep-pore volume (DPV) caused by annular lipids, a scenario consistent with our findings.

Our study differs in that we did not impose restraints to prevent lipid intrusion. By allowing lipids to interact freely with the pore, we systematically investigated their role in hydrophobic gating. Our results indicate that in the absence of lipids, the pore remains consistently hydrated, suggesting that lipid intrusion through lateral fenestrations is essential for hydrophobic gating in BK channels.

Furthermore, structural evidence supports our computational findings. Recent cryo-electron microscopy (cryo-EM) structures have shown lipid molecules occupying the fenestration cavities of BK channels, highlighting the significance of lipid interactions in channel gating.

We believe our study provides a more comprehensive understanding of the role of lipids in BK channel gating, reconciling observations from previous studies and offering new insights into the gating mechanism.

The rest of the study is quite thin. As such, I have a difficult time finding enough strengths in this work to reach a favorable recommendation for its publication in Nature Communications.

Thank you for your candid feedback. We are sorry to hear that you find the rest of our study lacking in depth. However, we would like to emphasize that our work represents, to the best of our knowledge, the most comprehensive investigation of BK channel pore dewetting to date. By simulating all the functional states currently available—Ca²⁺-free, EDTA-free, and Ca²⁺-bound—we provide a thorough analysis of how lipid intrusion through lateral fenestrations influences hydrophobic gating.

Our study not only explores the correlation between lipid occupancy and hydration free energy but also delves into the structural dynamics of the BK channel in various states. We believe that this comprehensive approach offers significant insights into the gating mechanisms of BK channels, which could have broader implications for understanding ion channel function.

1) About half of the paper is dedicated to analysis of water and ion distributions in Ca bound and free states. These properties are well documented in multiple studies at this point, including Jia et al 2018, Gu and de Groot 2023, Nordquist et al 2023, and Contreras et al 2024 (<https://doi.org/10.1101/2023.12.29.573674>).

Thank you for your insightful feedback. We understand your concern that a significant portion of our original manuscript was dedicated to the analysis of water and ion distributions in Ca²⁺-bound and Ca²⁺-free states, which have been well documented in previous studies such as Jia et al. 2018, Gu and de Groot 2023, Nordquist et al. 2023, and Contreras et al. 2024.

In response to your critique, we have substantially revised the manuscript to focus more on the novel aspects of our work, particularly the role of lipid intrusion in the hydrophobic gating of BK channels. Here are the key updates we have made:

- **Figure 1** now provides a comprehensive graphical overview of the three structures analyzed, clearly illustrating the presence and absence of lateral fenestrations in each structure. We have also included the newly available structure 8ghg to enhance the comparative analysis.

- **Figure 2** highlights the novelty of our work by offering new insights into the importance of lipid intrusion within the BK channel's inner cavity. This figure underscores how lipid penetration contributes to the hydrophobic gating mechanism.

- **Figure 3** presents novel results on the lipid intrusion process, utilizing metadynamics simulation data to shed light on the dynamics and energetics of lipid entry into the pore.

- **Figure 4** has been updated with additional data on hydration free energy and the metastability of various channel structures in the complete absence of lipids. This strengthens our argument about the essential role of lipids in pore dewetting.

- **Figure 5** reports on the effects of two specific mutations on lipid intrusion and pore hydration. These findings provide deeper insight into how structural modifications can influence lipid interaction and gating behavior.

- **Figure 6** illustrates crucial interactions between the protein and lipids, suggesting an allosteric coupling mechanism during the channel's opening and closing processes. This figure supports the idea that lipid interactions are not merely incidental but play an integral role in channel function.

To streamline the manuscript and focus on these novel contributions, we have moved some of the previously reported analyses of water and ion distributions in Ca^{2+} -bound and Ca^{2+} -free states to the **Supplementary Information**. We have also included new trajectory replicas in the supplementary materials to enhance the robustness and reproducibility of our results.

We believe that these revisions address your concerns by reducing redundancy with existing literature and emphasizing the unique findings of our study. By concentrating on the role of lipid intrusion and its impact on hydrophobic gating, we aim to contribute new and significant insights to the field.

2) Lipid fenestration in Ca-free BK channels is also well documented in both the above-mentioned simulation works, as well as a few published cryo-EM structures. The observation at this point from these studies is that lipid fenestration is inconsequential, as the pore was observed to be fully capable of dewet even without lipids in the pore. For example, a previous calculation of the pore hydration free energy profiles (Nordquist BJ 2023) show that the presence of lipid tail is not required for dewetting transition (in this work, restraints were imposed to exclude lipids from the inner cavity to achieve good convergence on the hydration free energy profiles). It is hard to speculate why the results reported in the current manuscript contradict all these previous simulations.

We appreciate the reviewer bringing up the previous studies and observations regarding lipid fenestration in Ca^{2+} -free BK channels. We acknowledge that lipid fenestration has been documented in earlier simulation works and cryo-EM structures. The reviewer notes that these studies suggest lipid fenestration is inconsequential to pore dewetting, as the pore can dewet even without lipids present.

However, we believe the apparent contradiction arises from methodological differences between our study and the previous ones, particularly regarding the treatment of lipid intrusion into the pore. In the work by Nordquist et al. (2023), the authors observed that lipid intrusion hindered the convergence of hydration free energy profiles. To address this, they applied restraints to prevent lipid tails from entering the pore's inner cavity. Specifically, they imposed a linear penalty on the lipid count within a spherical region of radius 5 Å to prevent lipids from penetrating deep into the pore region:

"To prevent these rare events and achieve better convergence of the pore hydration sampling, we use a similar 'coordination' CV to impose a restraint for preventing a lipid tail group from entering the pore. A linear penalty was applied to the lipid count [...] to prevent lipids from penetrating deep into the pore region." (Nordquist et al., 2023) Figure 2 from Nordquist et al., 2023 (reproduced here for the reviewer's convenience) shows how annular lipids are, in fact, inside the blue region of the DPV. Our results are not in contrast with these: we quantitatively characterize how the presence of these lipids affect the hydration properties of the BK pore.

[Figure Redacted]

One probably has to carefully inspect the structure integrity of the channel used in free energy calculations. One notable observation is that the human BK channel structure 6v3g was used in the simulation for Ca-free state. This structure contains a spurious pi-bulge in pore-lining helices. This feature is not present in any other BK structures reported so far; it is not consistent with McKinnon's own cryo-EM density map either. Nordquist et al examined this feature in depth. It was observed that 6v3g can be unstable in simulations and the pi bulge will dissolve back into regular alpha-helix after long relaxation.

Thank you for bringing up this important point regarding the structural integrity of the BK channel used in our free energy calculations. We note that Nordquist et al. reported lipid intrusion through the lateral fenestrations. This suggests that the presence or absence of the pi-bulge does not significantly influence lipid entry into the pore. In fact, in the supplementary materials of Nordquist et al. (Supp. Fig. S1, S2, S3), the authors not only show that lipids still enter the pore after equilibration, but also that there is a noticeable anticorrelation between lipid presence and water occupancy—an observation that aligns with our results.

I think it is imperative that the authors carefully examine the structures used in their calculations and reexamine the requirement of lipid fenestration in dewetting transitions.

Thank you for underscoring the importance of thoroughly examining the structural models used in our calculations and re-evaluating the necessity of lipid fenestration in dewetting transitions. We have taken your recommendation seriously and have expanded our structural analysis to address this concern. In particular, we have specifically investigated the role of the S6 helix kink angle in either permitting or preventing lipid entrance into the deep pore volume (DPV).

3) The definition of the DPV is potentially problematic as it only includes space where the lipid tail(s) would occupy. As such, it could exaggerate the correlation between pore hydration and lipid occupancy. Instead, one should define the entire inner pore volume, between the selectivity filter and cytosolic entrance, to examine how pore hydration depends on lipid occupancy.

Thank you for highlighting this important point about the definition of the deep pore volume (DPV). We addressed this concern by considering a larger volume for the calculation of hydration free energy profiles, as presented in Figure 4. Specifically, we included a larger volume encompassing not only the DPV but also the first ring of hydrophilic residues surrounding the hydrophobic core region of the pore, namely Thr287 and Glu321, as detailed in the Methods section. Our simulations revealed that the DPV is consistently the first area to undergo dewetting, while the surrounding regions of the pore cavity remain hydrated. This finding indicates that hydrophobic gating predominantly occurs within the DPV. Consequently, our analysis focused on this region, which is also reflected in the snapshots presented in Figure 4.

We apologize for any confusion arising from our initial definition of the DPV. To clarify this in the revised manuscript, we have added a detailed explanation in the Methods section and made slight modifications to Figure 4. These changes aim to better illustrate the volume considered in our analysis and to ensure that the correlation between pore hydration and lipid occupancy is accurately represented.

4) I am not sure the number of water near the bottom of the filter is a terribly meaningful indicator of the conductivity. Instead, it is the total number of water in the pore (or the level of dewetting in the entire pore) that is the most accurate indicator of the conductive state of the pore.

Thank you for bringing up this critical aspect of our study. We agree that the total number of water molecules within the entire pore lumen is an important indicator of the conductive state of the channel. However, it has been well established in the context of hydrophobic gating that dewetting of even a small hydrophobic region within a pore can create a significant free energy barrier to ion permeation. This localized dewetting can lead to complete suppression of ionic conductance due to the high energetic cost associated with ions traversing a dewet region (Zhu & Hummer, *Biophysical Journal*, 2012; Seiferth et al., *Journal of General Physiology*, 2022; Beckstein et al., *Journal of the American Chemical Society*, 2004).

In our investigation, we did not focus solely on the region near the bottom of the selectivity filter but considered the dewetting process throughout the entire pore lumen. Our simulations show that while the pore remains mostly hydrated, the area adjacent to the selectivity filter is the most susceptible to stable dewetting. This localized dewetting could play a crucial role in the hydrophobic gating mechanism of the BK channel by introducing a substantial energy barrier to ion conduction, even if the rest of the pore is hydrated.

5) the assertion that “the BK PGD volume seems to exceed the critical radius, length and hydrophobicity demonstrated to be needed to allow and maintain a bubble nucleation required to disrupt ion permeation” in the introduction does not seem to be correct. Previous studies of carbon nanotubes or idealized hydrophobic nanopores suggest a critical diameter of ~1 nm, which also depends on length of the pore and actual hydrophobicity. It is difficult to anticipate what “the critical radius, length and hydrophobicity” would be for a stable dewet BK pore. Nonetheless, the dewet inner pore of BK appears to be no greater than ~6Å in diameter (see Fig 2 or in previous simulation works (Jia et al 2018 and Gu and de Groot 2023)).

Thank you for highlighting this critical point regarding the dimensions and properties required for stable dewetting in hydrophobic nanopores. We acknowledge that previous studies on carbon nanotubes and idealized hydrophobic pores suggest a critical diameter of approximately 1 nm for stable dewetting, with the exact value depending on factors such as pore length and hydrophobicity.

Our original assertion was based on literature focusing on biological ion channels, where the contact angles between water and protein surfaces typically do not reach the high values observed in idealized systems (e.g., contact angles of 115°). Additionally, the hydrophobic regions in biological pores are generally shorter than those in carbon nanotubes, which can influence the dewetting behavior.

In the studies you mentioned, such as Rao et al. (*PNAS*, 2019), it is rarely reported that pores with a radius of around 0.3 nm can accommodate a stable vapor bubble. Specifically, their Figures 3 and 4, along with Supplementary Table S1, provide heuristic calculations of the water intrusion barrier ($\Delta G_w = 0$) for the Slo1 channel from *Aplysia Californica* (PDB ID: 5TJI). This structure, used by Jia et al. (2018) for homology modeling template of the human BK channel, shares a similar hydrophobic cavity with 6V3G and was predicted to be hydrated rather than dewetted. Furthermore, Jia et al. (2018) stated:

“The deep-pore region of hBK channels, lined by multiple nonpolar residues (L312, A313, F315, and A316), provide a baseline for the pore hydrophobicity, even though itself appears inadequate to trigger dewetting transitions.”

This suggests that the inherent hydrophobicity and dimensions of the BK pore may not be sufficient to induce stable dewetting without additional factors.

Indeed, the critical parameters for dewetting depend not only on pore radius but also on pore length and hydrophobicity. Theoretical studies have shown that longer hydrophobic regions promote more stable vapor bubbles (Giacomello and Roth, *Advances in Physics: X*, 2020; Paulo et al., *Nature Communications*, 2023). In biological pores like BK channels, these conditions are often not met due to shorter hydrophobic sections and moderate hydrophobicity.

Our hydration free energy calculations for the 6V3G pore without intruded lipids, as expanded upon in the revised Figure 4, support this understanding. The data indicate that the BK pore structure alone, without lipid intrusion, does not readily accommodate a stable vapor bubble, aligning with theoretical predictions and previous empirical findings.

6) the statement “Indeed, despite previous studies have shown a correlation between the dehydration level of the pore and protein structural features, e.g., the orientation of Phe residues in the central cavity or the radius shrinking [22], the complete dewetting of the region just below the SF was rarely reported” is confusing

Thank you for bringing this to our attention and we apologize for any confusion caused by our statement. We have rewritten the entire section to improve clarity and better compare our main results—specifically regarding the presence and role of lipids—with the previous literature taking into account all your previous comments.

Reviewer #2 (Remarks to the Author):

This manuscript uses MD simulations to study the gating mechanism of MK potassium channels, addressing the question of how these channels are gated in the absence of a pore occluding bundle crossing gate. They find that lipids penetrating through the lateral fenestrations are critical for gating, with the presence of lipid tails in the pore dehydrating the pore cavity (but not occluding it) preventing ion conduction. The result is genuinely interesting and intriguing, presenting a very unusual mechanism of ion channel gating and as such makes this content suitable for Nature Communications. But, it is essential to make sure the observation is statically significant prior to publication. As such I have a few suggestions for improvement.

Major concerns

1. The most interesting finding to me is that lipid tails in the pore generate pore dehydration which prevents ion conduction. Furthermore lipid tails only enter the pore in the Ca²⁺ free WT conformation, but not in the Ca²⁺ bound conformation or the Ca-free L312A or L312D mutants. Given the centrality of this results, I would like a more complete explanation as to why lipid tails don't enter the pore, as it is not obvious for the L312A mutant for example. In that case, the mutation is very subtle and would not be expected to occlude the fenestration, not does it have a big influence on hydrophobicity. SO why don't lipid tails enter in this case?

Thank you for your valuable feedback and for pointing out the need for a more detailed explanation regarding why lipid tails do not enter the pore in the L312A and L312D mutants. We appreciate your interest in our findings, and we agree that this aspect is central to understanding the gating mechanism we propose. Our simulations reveal that the L312A and L312D mutations induce structural changes that cause the channel to adopt conformations resembling the Ca²⁺-bound state. These changes have significant implications for lipid intrusion and pore hydration. The mutations lead to an expansion of the pore lumen in the deep pore volume (DPV) region. This is evident from the pore radius profiles presented in Figure 5c. The mutations cause a change in the tilt angle of the S6 helices, as shown in Figure 5b. The helices tilt outward, which reduces the size of the lateral fenestrations.

2. Following on from this, the lipid and water occupancy is determined from single 1 us simulations. It looks like to me that the authors only conducted one simulation for each condition (w/wo Ca²⁺) for 1us, in order to obtain statistical significance to show the solid difference observed between simulations, the authors must conduct multiple replicas. The authors can also make better use of the multiple fenestrations in each simulation that yield pseudo replicates that can better determine the uncertainty in the lipid occupancies.

Thank you for your insightful comment regarding the statistical significance of our simulations. We agree that conducting multiple replicas is essential to ensure the robustness and reliability of our results. We have indeed performed multiple replica simulations for each condition to address this concern and reported the results in the Supplementary material. The results from the replica simulations are consistent and support our original findings: lipid penetration occur in 2 of 3 collected trajectories for

the Ca²⁺-free structure 6V3G, while they are not observed in the 6V38 nor the 8GHG ones.

3. Finally, how much is the lipid occupancy dependent on the starting configurations of the lipids? The authors make use of charm-gui to prepare the systems but this doesn't do particularly well in placing lipids around the transmembrane protein with complex conformation like BK. If doing multiple replicates, I would suggest making use of multiple starting lipid coordinates as well.

Thank you for bringing up this important consideration regarding the dependency of lipid occupancy on the starting configurations of the lipids. We agree that the initial placement of lipids can influence simulation outcomes, especially for complex transmembrane proteins like the BK channel. Ensuring that our observations are not artifacts of initial conditions is crucial for the robustness of our conclusions. To address this concern, we have conducted multiple independent replica simulations (as mentioned above). For each replica, we used CHARMM-GUI to generate the system anew, allowing the software's stochastic algorithms to place lipids around the protein differently each time. This resulted in distinct initial lipid arrangements across replicas.

4. I would like additional justification for the choice of starting cryo-EM structures 6V38/6V3G which have relatively poor resolution, while there have been other BK channel structures in cell-derived membrane environment available (8GHF, 8GHG).

Thank you for bringing up this important point regarding our choice of starting cryo-EM structures. We appreciate your suggestion to consider more recent BK channel structures obtained from cell-derived membrane environments, such as 8GHF and 8GHG. In response to your critique and to strengthen our study, we have expanded our investigation to include the recently published structure **8GHG**. The inclusion of 8GHG, which represents a Ca²⁺-free and EDTA-free state of the BK channel, allows us to explore an additional conformational state and assess the generality of our findings.

Upon analyzing the simulations based on the 8GHG structure, we observed that lipid tails did **not** penetrate the pore cavity through the lateral fenestrations. Consequently, we did not detect dewetting induced by lipid intrusion in this structure. To further investigate this observation, we performed enhanced sampling simulations using metadynamics to explore the potential free energy landscape associated with lipid penetration. Our metadynamics simulations revealed that the free energy minimum corresponds to states with low lipid occupancy in the pore, which are insufficient to cause dewetting. This suggests that, in the 8GHG structure, the channel conformation and fenestration geometry do not favor lipid intrusion as a mechanism for hydrophobic gating.

While structures 6V38 and 6V3G have relatively lower resolutions compared to more recent models, we are confident that our findings regarding lipids occupying the lateral fenestrations are accurate and not artifacts of the structural data. This confidence is reinforced by a recent study by Kallure et al. (2023), currently available on bioRxiv, which provides clear and direct experimental evidence supporting our claim. In their Figure 2A (reproduced here for the reviewer's convenience), they explicitly demonstrate lipid molecules occupying the fenestration cavities of the BK channel. This observation corroborates our simulation results and validates the use of these structures in our study.

[Figure Redacted]

Kallure GS, Pal K, Zhou Y, Lingle CJ, Chowdhury S. High-resolution structures illuminate key principles underlying voltage and LRRC26 regulation of Slo1 channels. 2023 Dec 20:2023.12.20.572542. doi: 10.1101/2023.12.20.572542.

5. It is stated that the fenestrations are not present in the Ca²⁺ bound conformation. Can you give some analysis to support this? This can be done on the cryo-EM structures and ideally could include all the available BK cryo-EM structures to show it is consistent across structures with the same Ca²⁺ state.

To better illustrate the differences in fenestration among the analyzed structures, we have added surface volume representations for each structure in **Figure 1**. Additionally, we provide a quantitative assessment by including the distribution of the S6 helix tilt angles in **Figure 2**, as the bending of the S6 helices plays a crucial role in filling the fenestrations. While these results were previously presented in the Supplementary Information, we have now incorporated them into the main text in response to the reviewer's suggestion to highlight the changes in the S6 helices from the Ca²⁺-free to the Ca²⁺-bound states and the corresponding reduction in fenestration size.

Specifically, an increase in the tilt angles of the S6 helices results in narrower fenestrations. In the L312A and L312D mutants, although there is a shift in the S6 helix tilt angles toward the values observed in the Ca²⁺-bound state, this shift is not sufficient to completely fill the fenestrations and entirely block lipid entry. However, it does reduce lipid penetration below the threshold needed to induce dewetting.

6. The RMD results shown in Fig 4 are interesting. But I would also like to see analysis of the pore radius in each case to see if the lipids are occluding the pore or dehydrating it, as these correspond to distinct gating mechanisms.

We added a new Supplementary Figure S11 reporting the bottom view of the snapshots displayed in Fig.4c, that shows that lipids are not occluding the pore cavity. We also reported the analysis of the pore radius of that configuration in new Supplementary Fig. S12.

Minor comments:

7. When citing the early papers on hydrophobic gating I recommend you cite one of the first examples in biological channels: *Biophys. J.* 90: 799-810, 2006. I also recommend citing prior examples of lipid induced gating eg *Nature Commun* (2022) 13:490

Thank you for bringing these significant references to our attention. We have added a reference to the study by Ben Corry (*Biophys. J.* 90:799–810) and a reference to the recent publication by Jin et al. in *Nature Communications* (2022, 13:490).

8. Were any restraints used to keep the backbone restrained - and to keep Ca^{2+} free and Ca^{2+} bound conformations distinct? None are mentioned, but I just want to be sure.

Thank you for bringing this important point to our attention. We apologize for any confusion caused by the lack of detail in the Methods section. We have revised the section to clarify that **no backbone restraints were applied during the molecular dynamics (MD) simulations or the metadynamics simulations**. This means that the protein was free to move naturally, allowing us to observe intrinsic conformational changes without artificial constraints. In the case of the RMD calculations instead, we did apply restraints to the backbone atoms to characterize the wettability of well-defined conformational states.

9. It is stated that water/ion in SF are restrained. Why is this done and is it just for equilibration or in the production runs? This is particularly important as the authors were discussing the ion occupancy later, the occupancy of ion in SF has big impact of ion occupancy in pore cavity.

Thank you once again for pointing out the potentially confusing wording in our methods section. To clarify, we restrained only the two ions in the selectivity filter (SF) as well as the two resident water molecules throughout the entire simulation. We also explored the effect of the ion configuration by simulating two different SF setups: K^+ ions positioned in either the S1/S3 or the S2/S4 sites. Our results indicate that both configurations of the SF yield the same wetting properties for the cavity.

10. For the pore radii profile, please make it clearer when this is made using protein only and protein+lipid

We have clarified the distinction between the pore radius profiles calculated using only the protein and those calculated with both the protein and lipids. In the revised manuscript, we have retained the PDB ID as the label for the protein structures. For cases where the radius calculations include both the protein and lipids, we have explicitly labeled this as 'PDB ID + lipids' in the figure legend for Figure 2.

11. Fig 3B – I recommend making the labels clearer to indicate that the grey lines are water occupancy and the colours are lipid. For example the labels could be the same colour as the line and the flat colour lines could be made more visible (eg thicker lines)

The data previously shown in Figures 3B and 2B have been combined into Figure 2A in the revised manuscript. This change was made to better highlight the anticorrelation between lipid and water occupancy. We have maintained the color coding for the different structures and differentiated lipid and water counts by using lighter lines for lipids and darker dots for water. Additionally, in response to your suggestion, we have adjusted the line thickness and labels for improved visibility, making it easier to distinguish between the two components.

12. Please provide a reference when saying “These results are consistent with the experimental observation that the G-V curve of L321A in absence of Ca²⁺ is significantly left-shifted whereas L312D lead to constitutively open channel”.

Thank you for bringing this to our attention. We have revised the text and added a reference to the electrophysiological work by X. Chen, J. Yan, and R.W. Aldrich (Proc. Natl. Acad. Sci. U.S.A., 111(1), E79-E88, 2014).

Reviewer #3 (Remarks to the Author):

The authors describe a computational study that proposes a lipid mediated hydrophobic gating in the BK potassium channel. Hydrophobic/dewetting gating transitions have been proposed before in BK channels but the lipid involvement is for the first time systematically investigated here. Although in principle interesting, I have a number of concerns that should be addressed before I can recommend the manuscript for publication.

First, one of the main conclusions of the work "starting from a Ca²⁺-free leads to ion conduction upon Ca²⁺ binding." and "Ca²⁺ binding [...] allowing for pore hydration and conduction." is not backed up by the data, as no conductance has been demonstrated. The authors should show sustained currents in the calcium bound form consistent with experiments to support this claim.

Thank you for your feedback and for pointing out the ambiguity regarding our mention of conductance. We acknowledge that our study does not directly calculate conductance, but instead focuses on the hydration state, which is a key factor in ion conduction. To avoid confusion, we have revised the manuscript: In the abstract, we have rephrased the sentence to clearly reflect our findings without implying direct conductance calculations. In the main text, we emphasize the role of lipids in mediating the hydrophobic gating mechanism, noting that their absence leads to hydrated states, even in Ca-free structures.

Second, the presumed conductive state remains unclear in the current manuscript. In Figures 1 and 3, the selectivity filter is depicted as containing both water and ions in its binding sites. This is consistent with the description in the methods section: "Two potassium ions and two water molecules are allocated in the selectivity filter (TVGYG), in position S1-S3 and S2-S4 respectively.". However, in Fig. 4 filter configurations are shown containing 3 ions and no waters. The conductance for ion-only permeation might be quite different from a mechanism of ion/water co-permeation, so this should be clarified.

Thank you for pointing this out. The SF conformation (K-water-K-water) is maintained in the various analyses present in the study. In Fig. 4, the only water molecules shown are the one within the cavity below the SF.

Third, it is not explicitly stated which calcium parameters were used. Classical parameters that come with the employed force fields have been shown to show significant overbinding for divalent cations. Hence, specific multisite calcium parameters have been developed to mitigate this issue. Were these used in the current study?

Thank you for emphasizing the importance of the Ca²⁺ forcefield choice. Initially, we used the classical calcium parameter; however, in response to the reviewer's suggestion, we switched to the multisite calcium parameters for our replica simulations. Our updated analysis yielded results consistent with the original findings.

Finally, the manuscript suffers from numerous minor language issues and erroneous or missing literature references. To name just a few examples:

- the charmm36 reference is inaccurate
- Gromacs and plumed were mentioned but not cited
- "The two model have reproduce differently"
- "starting from a Ca²⁺-free leads"  "starting from a Ca²⁺-free state leads"?
- "we do not observed"

Thank you for your careful reading of the manuscript and your attention to all these crucial details. In addition to correcting the grammatical errors highlighted, we revised the citation for CHARMM36, now referencing the sources recommended on the CHARMM-GUI website. We also ensured that GROMACS and PLUMED are properly cited in the Methods section.

RESPONSE LETTER TO REVIEWERS

Reviewer comments are in italic, our replies are in red

Reviewer #1 (Remarks to the Author)

While I appreciate the efforts of the authors in responding to the concerns, I am unconvinced that my main concerns have been adequately addressed (reliability of the simulation, such as due to artifacts and instability problem of 6V38/6V3G; contrasting observations about the requirement of lipid fenestration for dehydration; limited new insights beyond the proposed role of lipids).

We appreciate Reviewer #1's continued engagement. However, we respectfully disagree with the assertion that our main conclusions are compromised by instability or lack novelty. In the absence of more detailed feedback, we briefly elaborate below on why we disagree with the reviewer.

On stability of 6V38/6V3G. We addressed this concern by conducting multiple independent replicas of our simulations to monitor the structural stability of protein. We now provide abundant evidence that neither the π -helix \rightarrow α -helix transition reported by Nordquist et al. [Biophysical Journal, Volume 122, Issue 7, 1158 - 1167], nor the occasional local backbone rearrangements resulted in the loss pore integrity or aberrant dehydration events. It is worth noting that this result is completely consistent with previous observations by Nordquist et al. [Biophysical Journal, Volume 122, Issue 7, 1158 - 1167] reporting that the structural transition in their simulation does not correlate with pore wetting (see Figs. S1–S2 therein). Taken together, these observations demonstrate that the lipid-induced dewetting is not the result of structural instability of the model as suggested by the reviewer. It is also important to emphasize that our simulations are based on the full-length protein, i.e. the most complete and accurate structural data available at present, comes from one of the most highly regarded groups in the field and, to our knowledge, has never been flagged as problematic for the pore conductance in the literature.

On contrasting observations about the requirement of lipid fenestration for dehydration. We respectfully disagree with the reviewer's interpretation—our conclusions do not contradict previously published findings. To clarify this point, we cite verbatim from the paper referenced by the reviewer (Nordquist et al., *Biophysical Journal*, Volume 122, Issue 7, pp. 1158–1167):

“To prevent these rare events and achieve better convergence of the pore hydration sampling, we use a similar ‘coordination’ CV to impose a restraint for preventing a lipid tail group from entering the pore. A linear penalty was applied to the lipid count [...] to prevent lipids from penetrating deep into the pore region.”

Our study addresses these rare events quantitatively: we computed well-converged hydration free energy profiles across various levels of lipid intrusion. This allowed us to quantitatively assess the interplay between the presence of annular lipids (entering through the fenestrations) and the transition from a “stable” wet state to a “stable” dry state.

On the novelty and impact of lipid-assisted gating. We disagree that our findings simply reiterate known mechanisms. To our knowledge, this is the first atomistic study to show how lipid tails entering fenestrations can actively bias the wetting/drying equilibrium of an ion channel pore – a mechanism that may well extend to other ion channels – as also recognized by the other reviewers. Our analysis is not only extensive and mechanistically detailed but also sets the stage for experimental testing. The enthusiastic reception of this work by the other reviewers further reinforces our belief in its relevance and impact.

Reviewer #2 (Remarks to the Author)

The authors have made considerable effort to address the comments in the first review including conducting a number of new simulations and analysis. I believe they have responded well to the comments I raised in my initial review. While lipid entry to the fenestrations and pore dewetting are both topics that have been discussed previously, the clear link between the presence of lipids in the fenestration and pore dewetting described here is novel and interesting in my opinion.

We thank Reviewer #2 for the positive assessment. We are delighted that our is seen as novel and interesting.

Reviewer #3 (Remarks to the Author)

The authors have satisfactorily addressed my concerns

We thank Reviewer #3 for the positive feedback. We are pleased that our revisions have satisfactorily addressed the reviewer’s concerns. We appreciate the helpful feedback and encouraging comments.